# Post-Harvest Red- and Far-Red-Light Irradiation and Low Temperature Induce the Accumulation of Carotenoids, Capsaicinoids, and Ascorbic Acid in *Capsicum annuum* L. Green Pepper Fruit

**DOI:** 10.3390/foods12081715

**Published:** 2023-04-20

**Authors:** Pavel Pashkovskiy, Nikolay Sleptsov, Mikhail Vereschagin, Vladimir Kreslavski, Natalia Rudometova, Pavel Sorokoumov, Aleksandr Ashikhmin, Maksim Bolshakov, Vladimir Kuznetsov

**Affiliations:** 1K.A. Timiryazev Institute of Plant Physiology, Russian Academy of Sciences, Botanicheskaya Street 35, Moscow 127276, Russia; mhlvrh@mail.ru (M.V.);; 2Department of Plant Physiology, Timiryazev Agricultural Academy—Russian State Agrarian University, Timiryazevskaya Street 49, Moscow 127434, Russia; inkss@mail.ru; 3Institute of Basic Biological Problems, Russian Academy of Sciences, Institutskaya Street 2, Pushchino 142290, Russia; vkreslav@rambler.ru (V.K.);; 4All-Russian Research Institute for Food Additives—Branch of VM Gorbatov Federal Research Center for Food Systems, Russian Academy of Sciences, St. Petersburg 191014, Russia

**Keywords:** *Capsicum annuum* fruit, red and far-red light, gene expression, biosynthesis of secondary metabolites

## Abstract

Environmental factors, such as light of different spectral compositions and temperature, can change the level of activated photoreceptors which, in turn, can affect the biosynthesis of secondary metabolites in the cells of green fruit. By briefly irradiating the harvested fruit of *Capsicum annuum* L. hot peppers with red light (RL, maximum 660 nm) and far-red light (FRL, maximum 730 nm) and by keeping them at a low temperature, we attempted to determine whether the state of phytochromes in fruit affects the biosynthesis of secondary metabolites. Using HPLC, we analysed the qualitative composition and quantitative content of the main carotenoids and alkaloids and the chlorophylls and ascorbate, in pepper fruit exposed to the above factors. We measured the parameters characterising the primary photochemical processes of photosynthesis and the transcript levels of genes encoding capsaicin biosynthesis enzymes. The total carotenoids content in the fruit increased most noticeably after 24 h of RL irradiation (more than 3.5 times compared to the initial value), and the most significant change in the composition of carotenoids occurred when the fruit was irradiated with FRL for 72 h. The capsaicin alkaloid content increased markedly after 72 h of FRL irradiation (more than 8 times compared to the initial value). It was suggested that decrease in the activity of phytochromes due to a low temperature or FRL may result in an increase in the expression of the *PAL* and *CAM* genes.

## 1. Introduction

Plants are rich in nutrients and fibre, and their regular use is necessary for humans. The consumption of fruits and vegetables reduces the risks of chronic and acute diseases such as cancer, obesity, and respiratory and cardiovascular diseases [1,2]. For this reason, new trends in the agricultural and horticultural industries are focused on exploring new ways to increase the content of bioactive compounds in foods. Plants of the Solanaceous family are an important component of dietary nutrition due to their high contents of flavonoids, carotenoids, and the alkaloid capsaicin. Capsaicinoids and some carotenoids have anticancer properties [3]. The fruits of *Capsicum annuum* are an important source of carotenoids, flavonoids, and vitamin C, which can slow the ageing process and serve as a preventive measure for a number of diseases [4].

During storage, the pigments in pepper fruits undergo changes which can have varying effects on their nutritional value. While chlorophyll levels typically decrease during ripening, the content of carotenoids, such as beta-carotene and lutein, increases, contributing positively to the nutritional profile of the fruit [5,6]. However, extended storage periods may lead to the degradation of certain pigments, potentially reducing the overall nutritional value of the pepper fruit [7]. As the fruit matures or ripens, the contents of lutein, violaxanthin, lycopene, β-carotene, capsanthin and/or capsorubin and other pigments change, what gives the pepper fruit an orange or red colour [8]. 

Some works indicate that environmental conditions can influence the rate of accumulation or degradation of essential nutrients such as carotenoids in fruits and vegetables during the processes of ripening and maturation [9,10]. 

One of the key environmental factors is light, which directly affects plant growth and the synthesis of primary and secondary metabolites [11,12]. By using light of different spectral compositions and intensities, it is possible to change the metabolic pathways for the formation of light-dependent secondary metabolites through the activation of certain photoreceptors [13]. Light controls plant development using a complex signalling cascade that involves transcription factors (TFs), kinases, calmodulin, ROS, and other components in addition to photoreceptors [14]. Light provides the differentiation of chloroplasts under changing conditions, which regulates photosynthetic activity. Some of the key photoreceptors are phytochromes (PHYs), which, in addition to the reception of red and far-red light (RL and FRL), are involved in the cross-signalling of blue light photoreceptors and are also temperature sensors [15,16,17,18]. Phytochromes have wide regulatory potential, and by binding to TFs such as phytochrome interacting factors (PIFs) and elongated hypocotyl 5 (HY5), they are able to regulate a complex of physiological processes, including the biosynthesis of various pigments [12,19]. FRL leads to a decrease in the content of the active form of PHYs, which reverses a cellular response by directing it along the opposite regulatory pathway [14,18]. In the dark, PHYs are inactive in the cytoplasm. When exposed to RL, a change in the structure of the chromophore leads to a rearrangement of the structure of apoproteins, which results in their translocation to the nucleus [20,21]. In the nucleus, PHY promotes the degradation of PIFs, which suppress, the expression of transcription factors such as HY5 in addition to other positive regulators of photomorphogenesis [22]. Changes in the activity of phytochromes significantly affect the content and activity of various components of phytochrome signalling, as well as the genes of antioxidant enzymes, photosynthetic proteins, and enzymes of the biosynthesis of low-molecular-weight antioxidants [23]. This leads to changes in the antioxidant status of plants, photosynthesis efficiency, and other metabolic responses [8,24]. It follows that light of different spectral compositions can be used to activate the expression of a complex of genes that affect change in the biosynthesis of vitamins, flavonoids, phenolic acids, carotenoids, and alkaloids [25,26]. For example, the combination of blue light and UV light increases the content of chlorophylls, carotenoids, capsaicin, phenols, and flavonoids in *Capsicum chinense* fruit [27]. RL has also been shown to accelerate the change in colour during wild-type tomato fruit ripening [12]. It has been suggested that this is due to the increased expression of ethylene biosynthesis genes. It was shown that supplementary RL increased the expression of genes encoding phytochromes in wild-type tomato fruit, which decreased the level of phytochrome-interacting factors, such as PIFs, increasing the activity of phytoene synthase and the content of phytoene. The reciprocal irradiation of tomato fruit with RL + FRL appears to be more effective in improving fruit quality than their separate treatment [11]. This indicates that PHYs may be involved in the processes of fruit ripening, including by changing the composition of carotenoids.

Temperature is another important environmental factor. It affects the metabolic activity of plant cells and consequently, the accumulation of secondary metabolites. In particular, a low temperature promotes the accumulation of anthocyanins while slowing cell growth [28,29,30]. Phytochromes are involved in temperature reception since the pool of the active forms of phytochromes decreases at low temperatures, similar to how it occurs under FRL [15,16].

Green pepper fruits have most of the characteristics of a green leaf since they contain photoreceptors, light signalling networks, and mature chloroplasts. In the cells of green fruit, the activity of photosystems I and II is preserved, as is as their ability to photosynthesize and form assimilates and secondary metabolites, although they are limited by the amount of minerals and carbohydrates stored before their separation from the mother plant, making them a convenient object for research. In addition, *C. annuum* plants are commonly cultivated crops with highly nutritional characteristics. However, the relationship between photosynthetic activity and the synthesis of secondary metabolites and the possible role of phytochromes in these processes have largely not been studied.

In connection with the above, the purpose of this work was to study how the activity of phytochromes adjusted using RL, FRL, and temperature can affect the qualitative and quantitative composition of carotenoids, alkaloids, and ascorbate, as well as the main photosynthetic processes and the expression of capsainoid biosynthesis genes in post-harvest green pepper fruit.

## 2. Materials and Methods

### 2.1. Plant Materials and Experimental Design

*C. annuum* L. (Korean long green) plants were grown until fruiting in a grow chamber with a 16 h photoperiod and at 25 °C, using fluorescent lamps (Philips, Pila, Poland) with an intensity of 250 µmol (photons) m^2^ s^−1^ in perlite and regular feeding with Hoagland nutrient medium. The fruits of this pepper are usually used fresh because they have a moderately spicy taste. Harvested green fruits were used for the experiment. The fruit were irradiated continuously from both sides (the light came from both sides simultaneously, and the fruits lay on the grid) under LED lamps with an intensity of 70 µmol (photons) m^2^ s^−1^ for 72 h. The light treatments used in the experiments were as follows: initial point—the beginning of the experiment (the same for all options); +4 °C darkness; +25 °C darkness; +25 °C light 660 nm 70 µmol (photons) m^2^ s^−1^; +25 °C far-red light 730 nm 70 µmol (photons) m^2^ s^−1^; +25 °C 660 nm 35 µmol (photons) m^2^ s^−1^ + 730 nm 35 µmol (photons) m^2^ s^−1^ (LEDs, Gorshkoff Prometheus series, Moscow Russia, www.gorshkoff.ru) (Figure 1). The LEDs were placed in fabric boxes with a foil coating measuring 1.5 by 1.5 m and 2 m high. The lamps were located at a distance of 30 cm from the fruit, on both sides, and the fruits were placed on a grid in such a way as to avoid it overheating. The temperature was stabilized by the general climate system and the exhaust fan system. The spectral characteristics of the light sources were determined using an AvaSpecULS2048CL-EVO spectrometer (Avantes, Apeldoorn, The Netherlands). Pepper fruit samples were fixed in liquid nitrogen at the beginning of the experiment and after 24 h and 72 h. In each version of the experiment, 20 pepper fruits were used with an average weight of about 50 g.

### 2.2. Pigment and Water Contents

The contents of the total chlorophylls (Chl *a* + *b*) and total carotenoids in the pigment extracts of the pepper fruits studied were determined spectroscopically in 80% acetone [31]. The total carotenoids content in the fruit pigment was obtained spectroscopically in 100% petroleum ether [32]. Each sample was measured in three biological replicates, and the contents of Chls and carotenoids were expressed as mg kg^−^ of dry weight.

The fresh weights of the fruits were determined with an accuracy of 1 mg using an analytical balance (Scout Pro SPU123, Ohaus Corporation, Parsippany, NJ, USA). The dry weight was determined using an analytical balance (AB54-S, Mettler Toledo, Greifensee, Switzerland) with an accuracy of 0.1 mg after drying the samples to a constant weight at 70 °C. The water content is expressed as a percentage of its fresh weight. Each sample was measured in three biological replicates.

### 2.3. HPLC Analysis of Carotenoids

A carotenoid analysis was performed as previously described [33]. Samples (200 mg) were homogenized in liquid nitrogen with pestle and mortar, and 10 mL of 100% acetone was then added to the resulting powder, stirred for 30 s, and incubated for 5 min at room temperature. Then petroleum ether (4 mL) was added and stirring was repeated. Next, 15–20 mL of distilled water was added, and the upper fraction of the petroleum ether was taken. The petroleum ether was dried under a stream of argon, and a film of carotenoids was then dissolved in 20 μL of acetone–methanol (7:2, *V*/*V*) and applied to an HPLC column. An HPLC device (Shimadzu, Kyoto, Japan) was assembled from the LC-10ADVP pump with an FCV-10ALVP module, a detector with a diode matrix, SPD-M20A, and a thermostat, the CTO-20 AC. The separation of the carotenoids was performed on a 4.6 × 250 mm reversed-phase column (Agilent Zorbax SB-C18, Agilent, Santa Clara, CA, USA) at 22 °C. The carotenoids were identified by their retention time and absorption spectra. The quantification of each carotenoid was performed by comparing its peak area in the region of 270–800 nm to the sum of all carotenoid peaks taken as 100% and was calculated with the LC-solution program ver. 1.0 (Shimadzu, Kyoto, Japan) using the molar extinction coefficients described elsewhere. Each sample was measured in three biological replicates.

### 2.4. HPLC Analysis of Ascorbate

Extraction was carried out using samples with an average weight of 200 mg. To achieve this, the powder ground in liquid nitrogen was immediately mixed with an extraction buffer of the following composition: 74.448 mg of EDTA; 286.65 mg of Tris-(2-carboxyethyl)-phosphine hydrochloride (TCEP), 5 mL of 98% orthophosphoric acid; Milli-Q water until 100 mL. The fresh extraction buffer was prepared each time immediately before analysis. The determination of the ascorbate content was performed as described earlier [34] using an HPLC system (Agilent 1100; Agilent Technologies, Santa Clara, CA, USA) equipped with a G1322A guart pump, a G1315B photodiode array detector, a G1322A degasser, and a 4.6 × 250 mm column (Waters spherisorb ODS2, 5 µm, Supelco Inc., Bellefonte, PA, USA). On the column, 20 µL of the sample was injected. For the elution of the ascorbate, the following gradient of solvents was used. An isocratic mobile phase A of 50 mM KH_2_PO_4_ (pH 2.5, adjusted with orthophosphoric acid) was applied for 5 min at a flow rate of 1 mL min^−1^, and then a short acetonitrile (mobile phase B) gradient from 0% to 30% was used between 3.5 and 6 min to elute the less polar components from the column. Between 8 and 9 min, the percentage of mobile phase B was decreased to 0%, and the column was equilibrated to mobile phase A for an additional 9 min. The total running time was 18 min. The analysis was carried out at room temperature. The signal was recorded between 190 and 500 nm. The ascorbate peak appeared at approximately 5.1 min with an absorption maximum at approximately 244 nm. A two-fold serial dilution between 0.2 and 50 µM ascorbate was dissolved in the extraction buffer as a standard. The peak areas of ascorbate in the HPLC chromatograms were determined by software using linear fitting to obtain a calibration curve. Each sample was measured in three biological replicates, and the content of ascorbate was expressed as mg kg^−1^ of dry weight.

### 2.5. HPLC Analysis of Capsaicinoids

The analysis was carried out using the method in [35] with some modifications. The analysis was carried out on a Varian 920-LC high-performance liquid chromatograph with a photodiode array detector. The separation of the sample components was carried out on an Agilent Hypersil 5 AA-ODS chromatographic column (2.1 × 200 mm). Pepper samples were homogenized in liquid nitrogen, and a 1 g sample of pepper was dried for 24 h in a lyophilizer (HyperCOOL HC3055, Seoul, Republic of Korea). The obtained dry powder (200 mg) was subjected to extraction in 3 mL of methanol, thoroughly shaken for 3 min, placed in an ultrasonic bath (Sapphire-2.8 TTC, Moscow, Russia) for 30 min at 40 °C (power 160 W), and shaken again for 3 min and filtered, after which the filter was washed with 5 mL of methanol. The obtained filtrate was concentrated in a vacuum centrifuge (Eppendorf Concentrator Plus, Hamburg, Germany) to dryness and dissolved in 200 µL of methanol. The resulting extract was introduced into the chromatograph with the following parameters: elution mode isocratic, eluent (methanol:acetonitrile (50:50)), column thermostat temperature: 35 °C; sample volume 20 µL; eluent flow rate: 0.3 mL min^−1^; analysis time: 12 min. Detection was performed by a photodiode array detector with the detection wavelengths λ_1_ = 280 nm and λ_2_ = 230 nm. To build a calibration dependence, solutions of capsaicinoid standards in methanol with concentrations of 1, 10, 50, and 100 µg mL^−1^ were prepared. The solutions were analysed sequentially according to the above analysis conditions, from the lowest concentration to the highest, with two parallel measurements of each standard solution. The retention time of capsaicin and nonivamide was 6.3 min, and the retention time of dihydrocapsaicin was 8.4 min. From the data obtained, calibration characteristics were calculated: C_cap+non_ = 0.554 S − 0.389, where C_cap+non_ is the total concentration of capsaicin and nonivamide in the extract, μg mL^−1^; S is the area of the signal on the chromatogram, with a retention time of 6.2 min. C_dhcap_ = 0.923 S + 0.202, where C_dhcap_ is the concentration of dihydrocapsaicin in the extract, μg mL^−1^; S is the area of the signal on the chromatogram, with a retention time of 8.2 min. The capsaicinoid content in the samples was calculated according to the following equations: for capsaicin and nonivamide C_1_ = ((C_cap+non_) · 0.2) m^−1^; for dihydrocapsaicin C_2_ = (C_dhcap_ · 0.2) m^−1^, where: C_1_ sample is the total concentration of capsaicin and nonivamide in the sample, µg g^−1^ dry weight; C_2_ is the concentration of dihydrocapsaicin in the sample, µg g^−1^ dry weight; C_cap+non_ is the concentration of capsaicin and nonivamide in the sample extract, μg mL^−1^; C_dhcap_ is the concentration of dihydrocapsaicin in the sample extract, µg mL^−1^; 0.2 is the extract volume, mL; m is the mass of the sample, g. Each sample was measured in three biological replicates, and the capsaicinoid content was expressed as mg kg^−1^ of dry weight.

### 2.6. Determination of Photochemical Activity

The fluorescence induction curves were measured with a mini-PAM II fluorometer (Walz, Germany, Effeltrich) on fruits adapted to the dark (30 min). After a pulse of saturating light, the fruits adapted to the dark for 30 min were kept in the dark for one minute. They were then exposed to actinic light for 5 min, followed by saturating light pulses during which the parameters were measured. Blue LEDs (450 nm) were used to provide the measuring light (0.5 μmol (photons) m^−2^ s^−1^), actinic light (150 μmol (photons) m^−2^ s^−1^), and saturating pulses (450 nm, 5000 μmol (photons) m^−2^ s^−1^ and 800 ms duration). The parameter calculations, calculated on the basis of the fluorescence data, were performed using WinCntrl v.2.41a software (Walz, Germany). The values for F_0_, F_v_, F_m_, F_m_’, and F_0_’, as well as the PSII maximum (F_v_/F_m_) and effective (Y(II) (F_m_’ − F_t_))/F_m_’ photochemical quantum yields, were determined. F_m_ and F_m_’ are the maximum Chl fluorescence levels under dark- and light-adapted conditions, respectively. F_v_ is the photoinduced change in fluorescence, and F_t_ is the level of fluorescence before a saturation impulse is applied. F_0_ is the initial Chl fluorescence level. The actinic light was switched on for 10 min (I = 125 µmol (photons) m^−2^ s^−1^).

### 2.7. RNA Extraction and RT-PCR

RNA isolation was performed according to a previously described method [36]. The quantity and quality of the total RNA were determined using a NanoDrop 2000 spectrophotometer (Thermo Fisher Scientific, Waltham, MA, USA). cDNA synthesis was performed using the M-MLV Reverse Transcriptase Kit (Fermentas, Burlington, ON, Canada) and the oligo (dT) 21 primer. The expression patterns of the genes were assessed using the CFX96 Touch™ Real-Time PCR Detection System (Bio-Rad, Berkeley, CA, USA). A real-time PCR was performed using the SYBR ^®^ Green qPCR supermixes for real-time PCR (Eurogen, Moscow, Russia); the primer’s melting temperature was 60 °C, using the comparative cycle threshold (∆∆Ct) method. The transcript levels were normalized to the expression of the *Actin1* gene. The level of gene transcripts at the initial point of the experiment was taken equal to 1. Gene-specific primers for *C. annuum* phenylalanine ammonia-lyase-like (*PAL*, NM_001324603.1) forward AAAGTGCCGAGCAACACAAC, reverse AAAGCGCCACGAGATAGGTT; *C. annuum* caffeoyl-CoA O-methyltransferase (*CAM*, NM_001324582.1) forward TCGCACAAGATTGGTGATGGT, reverse CACTGGTTGAGATCGGACAAA; *C. annuum* capsaicin synthase (*CSY1*, DQ349223.1) forward TGATCTTCATTTTGACCGTAAACTT, reverse CCTCTCCGGGTATTTCACCG; *C. annuum* acyl-CoA synthetase (*ACS*, AF354454.1) forward AGTCATCGTGCGATTTCCGT, reverse TTGGGGAAAGTGAGAGCGAC; *C. chinense* fatty-acid thioesterase acyl-ACP thioesterase (*FATa*, AF318288.1) forward AGGACTTGTGCCACGAAGAG, reverse TCATGCTGGCATTCACGTCT; *C. chinense* ketoacyl-ACP synthase I (*KAS*, AF085148.1) forward CTTGTTGCTGGGAAAAGGGC, reverse CTTTGCACCAAGGCTTCCAC, and *C. annuum* Actin (*Act1*, AY572427.1) forward TTAGCAACTGGGATGACATGGA reverse CCTGAATGGCAACATACATAGCA were selected using nucleotide sequences from the National Center for Biotechnology Information (NCBI) database (www.ncbi.nlm.nih.gov, USA) (accessed on 1 December 2022) with Vector NTI Suite 9 software (Invitrogen, Waltham, MA, USA). The experiments were performed with three biological and analytical replicates. 

### 2.8. Statistical Data Processing

The experiments were performed in three biological replicates and three analytical replicates. The expression level of each gene was measured in three independent experiments. For each of these experiments, at least three parallel independent measurements were performed. The significance of the differences among the groups was calculated by a one-way analysis of variance (ANOVA) followed by Duncan’s method, using SigmaPlot 12.3 (Systat Software Inc., Chicago, IL, USA). Small letters indicate significant differences between the initial point (0 h of experiment) and different light and temperature conditions (*p* < 0.05). The data are shown as the mean ± SD (*n* = 3).

## 3. Results

### 3.1. Carotenoid, Pigment, and Water Contents

The influence of different qualities of light and a low temperature led to a decrease in the content of chlorophyll relative to the initial point of the experiment. After 72 h of exposure, the most significant increase in the contents of carotenoids was observed in the RL + FRL variant (more than two times higher than in the initial point) (Table 1).

After 24 h of irradiation, a noticeable increase in the amount of carotenoids and xanthophylls (3.6 times higher than in the initial point) was observed in the RL variant (0.51 g kg^−1^), and after 72 h in the RL + FRL variant, the content had approximately tripled (Table 1). In variant RL + FRL, a decrease in the percentage of epoxy-lutein, zeaxanthin, and cis-mutatoxanthin, which was also accompanied by a decrease in the percentage of β-carotene, was observed (Table 2).

After 24 h of exposure, an increase in carotenoids was also observed under the FRL conditions (0.31); however, it was only two times higher than in the control. The difference between the FRL and the RL conditions was only that the increase in percentage was due to other carotenoids, namely, violaxanthin, and lutein. Exposure to a low temperature was also accompanied by a decrease in the percentage of β-carotene, but only after 72 h (Table 2). At 25 °C in the dark, the β-carotene percentage decreased compared to the initial point, probably due to an increase in the neoxanthin percentage (Table 2).

Over the course of the study, the percentage of water loss in the pepper fruit samples was estimated. In the first 24 h, water loss was the most significant in the RL + FRL and FRL variants, while in the dark and under RL, dehydration was less pronounced. The smallest water loss was detected in the samples maintained at a low temperature (0.6% per day). After 72 h, the greatest water losses were observed in the RL + FRL variant, and the smallest were observed at a low temperature (Table 1). 

### 3.2. Primary Photochemical Processes of Photosynthesis

The maximum quantum yield remained at a high level under the dark, FRL, and low-temperature conditions; however, in the 24 h and 72 h variants with RL and RL + FRL irradiation, a decrease in this index was observed which was most pronounced in the RL + FRL variant. The effective quantum yield Y(II) decreased in the first 24 h for the RL option compared with the initial point. However, under FRL and RL + FRL conditions after 24 h, Y(II) remained at the control level, with a decrease after 72 h. At the same time, RL and especially RL + FRL irradiation led to a noticeable decrease in the Y(II) index (0.12 and 0.08, respectively) (Table 1).

### 3.3. Ascorbate Content

The ascorbate content increased in all spectral options excluding the dark variant at 24 h when compared to initial value. Thus, after 24 h under the RL condition, the ascorbate content was higher than in the initial value by 1.5 times, and after 72 h at the dark and at low temperatures, it was higher than the initial value by 1.6 and 1.7 times, respectively (Figure 2c).

### 3.4. Content of Alkaloids

Over the course of the study, the contents of the main alkaloids of the pepper, namely, capsaicin, dehydrocapsaicin, and nonivamide, were determined. The largest increase in the nonivamide and capsaicin contents was observed after 72 h under the FRL condition. It is also important to note that at low temperatures, there was a significant increase in the contents of both of these capsaicinoids after 24 h and 72 h, while the smallest alkaloid contents were observed when keeping fruit in the dark at 25 °C (Figure 2).

### 3.5. Gene Expression

In this work, the expression of genes for the synthesis of capsaicin and the expression of phenylalanine ammonia-lyase, the key enzyme for the synthesis of phenylpropanoids, was studied. The expression of the gene caffeoyl-CoA O-methyltransferase (*CAM*) increased in the RL + FRL variant after 24 h of irradiation by more than three times relative to the control and remained increased until the end of the experiment, although the highest expression of this gene was observed in the FRL and RL variants for 72 h at eight and six times higher than the control, respectively. The transcript level of acyl-ACP thioesterase fatty-acid thioesterase (*FATa*) was maximal in the 24 h RL variant (more than six times higher than in the initial value) and in the +25 °C darkness variant at more than eight times higher than in the control. The transcript level of ketoacyl-ACP synthase (*KAS*) increased more than 10 times upon cold treatment for 24 h, as well as upon irradiation of the fruit with FRL for 72 h. At the same time, after 24 and 72 h, *KAS* expression increased by two or more times in the RL + FRL variants. After 24 h, *ACS* expression increased by almost 3-fold at low temperatures and remained at the same level until the end of the experiment. *PAL* expression increased 5-fold when the fruits were exposed to RL + FRL for 24 h, and it increased 6-fold when FRL was trained for 24 h relative to the control. The maximum increase in *PAL* expression was observed in the 72 h in the FRL option, which demonstrated an increase of more than 15-fold, and in the RL option, which demonstrated an increase of more than 12-fold relative to the control. Transcriptional activation of this gene was also observed in the RL + FRL variants in the darkness at 25 °C and 4 °C, although to a lesser extent (Figure 3).

## 4. Discussion

Due to its photosynthetic pigment contents and the functioning of its photoreceptors and light signalling networks, green fruit is able to maintain photosynthetic activity for a long time and behave like plant leaves. Under normal conditions, fruit ripening occurs spontaneously and uncontrollably. It is known that different varieties of hot peppers can react differently to light of various spectral compositions. While the effect of RL on the biosynthesis of secondary metabolites has been previously studied, the impact of FRL and the combination of far-red light and RL have not yet been sufficiently explored [37]. We attempted to understand how this process can be influenced by RL, FRL irradiation, and exposure to low temperature, which affects the content of the active form of phytochromes [18]. As phytochromes are active in green fruit, they are able to influence the activity of metabolic processes, as in leaves [12]. In our experiments, we observed a decrease in the chlorophyll content by almost 2-fold relative to the beginning of the experiment, and the most significant decrease was observed during irradiation with RL + FRL for 72 h (Table 1). Under irradiation with RL + FRL, a decrease in PSII activity was also observed, estimated from F_v_/F_m_ and Y(II) values, although Fv/Fm was lower than in leaves [24]. In addition to the Y(II) value, the F_v_/F_m_ also decreased under long-term exposure to RL and RL + FRL but not under exposure to FRL and a low temperature (Table 1), indicating that photosynthetic activity can be maintained at a high level under FRL irradiation. Apparently, this is because the presence of RL in the light spectrum leads to an acceleration of photosynthesis in cells and a faster consumption of the pool of assimilates, which ultimately leads to a decrease in chlorophylls and photosynthetic activity and, as a consequence, accelerated maturation.

Carotenoids transfer energy to chlorophylls for photosynthesis, protect them from photooxidation [38,39], and can serve as antioxidants. Similar to the chlorophyll content, the carotenoid content varied in different plants with an increase in the proportion of FRL [40,41]. In our experiments, the amount of carotenoids and xanthophylls increased under FRL after 24 h by two times relative to the initial value, while the largest amount of carotenoids was found in the RL variant (0.51 g kg^−1^), which was 3.6 times higher than the initial value (Table 1). Similar results under RL conditions were obtained in the work [42]. On the contrary, at a higher intensity of RL when illuminating the sweet peppers, a decrease in carotenoids was observed [43]. It can be assumed that the increase in the content of carotenoids occurred due to an increase in the percentage of zeaxanthin, epoxy-lutein, and cis-mutatoxanthin and was accompanied by a decrease in β-carotene (Table 2). Cis-carotenoids are red in colour and have a high antioxidant capacity [44]. Most likely, the increase in these carotenoids is associated with a need for cell photoprotection, which is additionally indicated by reduced photochemistry parameters, indirectly indicating the possible development of oxidative stress in cells (Table 1 and Table 2). In the leaf, a high content of the active form of phytochrome can prevent the loss of carotenoids, while FRL light can reduce the relative content of the active forms of phytochrome and carotenoids, acting through the PHY-PIFs-Car interaction [12,45]. In the pepper fruit, after 72 h of FRL irradiation, we observed a decrease in the content of carotenoids, which is consistent with the data obtained on the leaves (Table 1). At the same time, the irradiation of fruit with FRL led to an increase in the diversity of carotenoids and xanthophylls against the background of their general decrease (Table 2). An increase in the content of ascorbate, which is one of the most powerful nonenzymatic antioxidants, probably also indicates the need for RL-induced photoprotection. In the RL 24 h variant, the ascorbate content was 1.5 times higher than the ascorbate content in the control (Figure 2c). The ascorbate content was comparable when exposed to low temperature. This indicates that both the active and inactive forms of the phytochrome can influence the increase in ascorbate content.

Phytochromes are able to influence the accumulation of secondary metabolites in leaves both directly and through a network of light-sensitive TFs [12] which in turn, regulate the expression of genes for the biosynthesis of enzymes of the phenylpropanoid pathway, for example, phenylalanine ammonia-lyase *PAL* [46]. Capsaicinoids, the main secondary metabolites of the pepper, are synthesized in two ways, the products of which are combined at the end of biosynthesis. The first pathway is the phenylpropanoid pathway, the key enzyme of which is also *PAL*, in which cinnamic acid is formed from phenylalanine, followed by cinnamate-4-hydroxylase (C4H), 4-coumarate-coenzyme A-ligase (4CL), hydroxycinnamoyltransferase (HCT), coumarate-3-hydroxylase (C3H), caffeyl-CoA-3-O-methyltransferase (*CAM*), and aminotransferase (pAMT) to form vanillamine. In pepper fruit, we observed a 3-fold increase in *PAL* gene expression in the first 24 h after irradiation with FRL and RL + FRL (Figure 3). When the fruits were irradiated with RL + FRL, *CAM* gene expression was additionally increased by more than 3-fold relative to the initial value (Figure 3). In [47], it was shown that in addition to *PAL* and capsaicin synthase (*CAS*), the biosynthesis of fatty acid genes plays a decisive role in the biosynthesis of capsaicinoids. The second pathway for the formation of capsaicinoids is associated with the biosynthesis of fatty acids. It includes the enzymes ketoacyl-ACP synthase I (*KAS*), malonyl-CoA-ACP transacylase (MCAT), acyl transfer protein (ACL), and *FATa* thioesterase (TE), among others [47,48]. It is not known whether the genes for the synthesis of fatty acids *ACL1*, *FATa*, and *KAS*, which are necessary for the formation of capsaicinoids, can be regulated by phytochromes; however, it is known that the differential expression of these genes in pepper fruit positively correlates with the content of these alkaloids [47]. Notably, the highest expressions of the *CAM*, *ACS*, *KAS*, and *CAS* genes were observed at 4 °C during the first 24 h of the experiment (Figure 3). This indicates the possibility of the activation of the expression of these genes with the help of low temperature, when phytochromes are inactive forms. This is quite logical, since the activation of lipid biosynthesis under cold conditions has been described in many works [49,50]. The active form of a phytochrome spontaneously reverts back to its inactive form during thermal reversal, allowing phyB to act as a thermosensor [15,16]. Temperature, as well as FRL, can influence the transition of phytochromes from active to inactive forms [15,16]. This explains the decrease in phytochrome activity under conditions of a lower temperature, which had a positive effect on the expression of alkaloid biosynthesis genes and, in our experiments, led to an increase in the contents of capsaicin and dihydrocapsoicin at 4 °C (Figure 2a,b). At the same time, by the third day of the experiment, the highest expressions of *CAM*, *PAL*, *FATa*, *CSY*, and *KAS* were observed with FRL irradiation, which was accompanied by an increase in the content of capsaicin (Figure 2a). This circumstance additionally confirms the need for Pr to activate the biosynthesis of alkaloids. Another interesting observation was that in the RL + FRL variant, despite a lower level of gene transcription, the qualitative composition of capsaicinoids changed in favour of dihydrocapsaicin (Figure 2b), which was not observed in other variants.

## 5. Conclusions

The ripening of green pepper fruit can be controlled by changing the temperature and light conditions in narrow time ranges by changing the pool of the active form of phytochromes with the help of irradiation with RL, FRL, and exposure to a low temperature. Ripening is accompanied by a decrease in the content of chlorophylls and the photosynthetic activity of fruits, which leads to a change in the quantitative content and qualitative composition of carotenoids (an increase in the percentages of neochrome, neoxanthin, violaxanthin, capsanthin, capsorubin, anteroxanthin, and cis-mutatoxanthin but a decrease in zeaxanthin, lutein, neolutein, and β-carotene) and alkaloids. One possible reason for the observed phenomena could be an enhanced light-induced expression of the *PAL* gene and genes for the enzymes of alkaloid biosynthesis, such as *CAM* and *KAS*. At the same time, an increased alkaloid content was observed at a low temperature and under FRL, which indicates a decreased involvement of the active forms of phytochromes in the biosynthesis of alkaloids in green pepper fruit. This allows us to propose an efficient way to increase the content of valuable carotenoids and alkaloids through the short-term processing of harvested green fruit with RL and FRL immediately before its use in everyday life.

## Figures and Tables

**Figure 1 foods-12-01715-f001:**
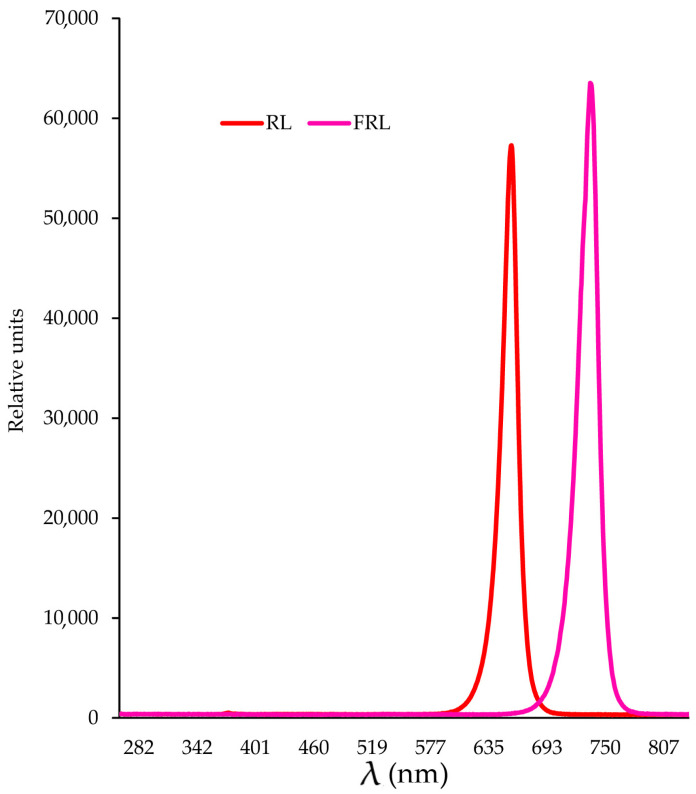
Emission spectra of the light sources used in the experiments.

**Figure 2 foods-12-01715-f002:**
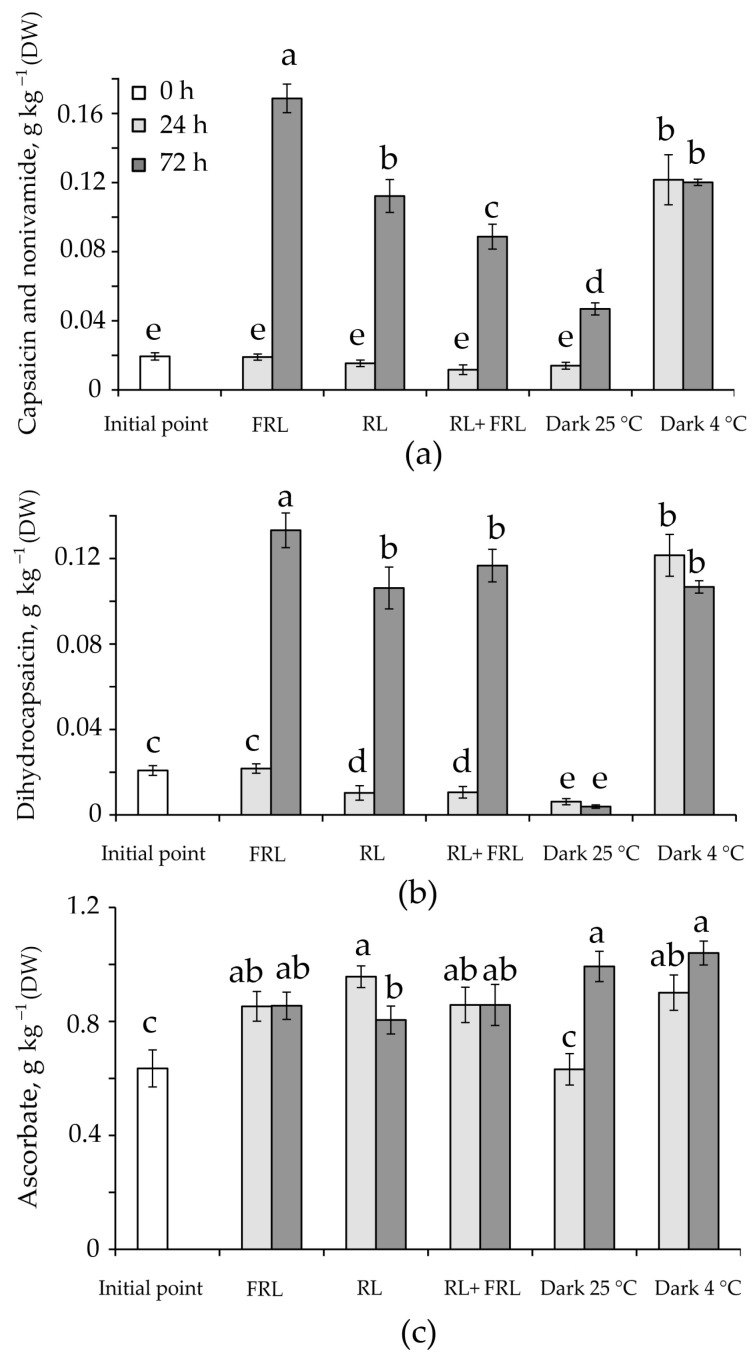
The content of the main capsaicinoids, capsaicin and nonivamide (**a**), dihydrocapsaicin (**b**), and ascorbate (**c**) under the influence of RL, FRL, and a low temperature in *C. annuum* fruit after 24 and 72 h. Different letters indicate significant differences (*p* < 0.05) between the experimental treatments. The means ± standard errors, *n* = 3.

**Figure 3 foods-12-01715-f003:**
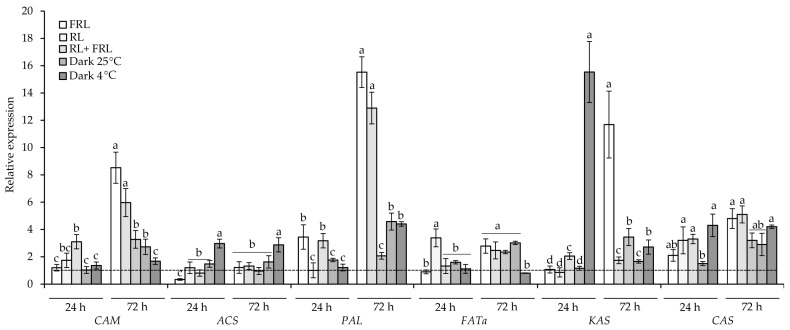
Transcript levels of caffeoyl-CoA 3-O-methyltransferase (*CAM*), acyl-CoA synthetase (*ACS*), phenylalanine ammonia lyase (*PAL*), fatty-acid thioesterase (*FATa*), ketoacyl-ACP synthase I (*KAS*), and capsaicin synthase (*CAS*) genes under RL, FRL and low-temperature conditions in *C. annuum* fruit after 24 and 72 h. The transcript levels were normalized to the expression of the *Actin1* gene. The gene expression at 0 h of the experiment (initial point) was used as one unit (dotted line). Different letters indicate significant differences (*p* < 0.05) between the experimental treatments for each gene. The means ± standard errors, *n* = 3.

**Table 1 foods-12-01715-t001:** The content of the sum of chlorophylls, carotenoids, and xanthophylls, water loss (%), and indicators of the maximum and effective quantum yield of PSII under the influence of RL, FRL, and low temperature in *C. annuum* fruit after 24 and 72 h.

	Time, (h)	Chl (*a* + *b*), g kg^−1^ (DW)	Car + Xan, g kg^−1^, (DW)	Y(II)	F_v_/F_m_	Water Loss, %
Initial point	0	14.3 ± 0.7 ^a^	0.14 ± 0.02 ^e^	0.22 ± 0.01 ^b^	0.687 ± 0.021 ^ab^	
FRL	24	11.2 ± 0.4 ^b^	0.31 ± 0.01 ^c^	0.22 ± 0.01 ^b^	0.663 ± 0.026 ^ab^	6.81 ± 0.61 ^ab^
RL		10.2 ± 0.3 ^bc^	0.51 ± 0.02 ^a^	0.15 ± 0.02 ^bc^	0.545 ± 0.023 ^c^	5.95 ± 0.42 ^b^
RL + FRL		11.8 ± 0.5 ^b^	0.29 ± 0.01 ^c^	0.20 ± 0.02 ^b^	0.454 ± 0.021 ^d^	8.21 ± 0.51 ^a^
Dark 25 °C		11.1 ± 0.6 ^b^	0.21 ± 0.01 ^d^	0.28 ± 0.02 ^a^	0.650 ± 0.021 ^b^	4.66 ± 0.21 ^c^
Dark 4 °C		10.4 ± 0.4 ^bc^	0.25 ± 0.01 ^bc^	0.23 ± 0.03 ^ab^	0.662 ± 0.022 ^ab^	1.54 ± 0.21 ^e^
FRL	72	11.6 ± 0.3 ^b^	0.19 ± 0.01 ^de^	0.20 ± 0.02 ^b^	0.655 ± 0.021 ^ab^	3.63 ± 0.05 ^d^
RL		12.8 ± 0.5 ^b^	0.32 ± 0.01 ^c^	0.12 ± 0.01 ^c^	0.322 ± 0.023 ^e^	4.02 ± 0.12 ^d^
RL + FRL		9.0 ± 0.3 ^c^	0.42 ± 0.02 ^b^	0.08 ± 0.01 ^d^	0.161 ± 0.023 ^e^	5.48 ± 0.04 ^bc^
Dark 25 °C		11.0 ± 0.5 ^b^	0.28 ± 0.01 ^c^	0.25 ± 0.02 ^ab^	0.736 ± 0.021 ^a^	4.72 ± 0.12 ^c^
Dark 4 °C		11.1 ± 0.5 ^b^	0.29 ± 0.01 ^c^	0.20 ± 0.01 ^b^	0.68 ± 0.024 ^ab^	1.52 ± 0.11 ^e^

Chl—chlorophyll; Car + Xan—carotenoids xanthophylls; DW—dry weight; Y(II)—effective quantum yield of PSII; Fv/Fm—maximum quantum yield of PSII. Different letters indicate significant differences (*p* < 0.05) between the experimental treatments. The means ± standard errors, *n* = 3.

**Table 2 foods-12-01715-t002:** The carotenoid compositions in percentages (%) of total content under the influence of RL, FRL and low temperature in *C. annuum* fruit after 24 and 72 h of the experiment.

Time (h)	0	24					72				
	Initial Point	FRL	RL	RL + FRL	Dark 25 °C	Dark 4 °C	FRL	RL	RL + FRL	Dark 25 °C	Dark 4 °C
Neoxanthin	8.31 ± 0.91 ^b^	8.81 ± 1.31 ^b^	6.11 ± 0.81 ^c^	11.12 ± 1.1 ^ab^	10.50 ± 1.12 ^ab^	8.31 ± 1.25 ^b^	13.61 ± 1.61 ^a^	12.41 ± 1.21 ^a^	9.81 ± 1.04 ^b^	8.75 ± 1.64 ^b^	9.20 ± 0.94 ^b^
Violoxanthin	18.01 ± 1.51 ^c^	28.12 ± 2.24 ^b^	24.12 ± 2.85 ^b^	17.16 ± 0.91 ^c^	20.30 ± 2.84 ^bc^	28.61 ± 2.35 ^b^	39.31 ± 3.24 ^a^	28.43 ± 2.84 ^b^	22.42 ± 2.85 ^b^	26.85 ± 2.72 ^b^	33.11 ± 1.24 ^b^
Neochrome	0	0	0	0	0	0	0.60 ± 0.12 ^a^	0	0	0	0
Capsanthin	0.11 ± 0.03 ^c^	0.21 ± 0.03 ^b^	0.10 ± 0.03 ^c^	0	0.20 ± 0.03 ^b^	0.21 ± 0.03 ^b^	0.50 ± 0.07 ^a^	0.11 ± 0.02 ^c^	0.11 ± 0.02 ^c^	0.35 ± 0.03 ^b^	0.21 ± 0.03 ^b^
Capsorubin	0.31 ± 0.05 ^b^	0.11 ± 0.05 ^d^	0	0.21 ± 0.04 ^c^	0.30 ± 0.04 ^bc^	0.40 ± 0.06 ^b^	0.91 ± 0.09 ^a^	0.50 ± 0.07 ^b^	0	0.40 ± 0.05 ^b^	0.21 ± 0.04 ^c^
Cis-mutatoxanthin	0	0	5.00 ± 1.85 ^a^	0	0	0	0.60 ± 0.35 ^b^	0	5.00 ± 0.26 ^a^	0	0
Anteroxanthin	0.31 ± 0.05 ^b^	0	0	0.11 ± 0.05 ^c^	0	0.10 ± 0.05 ^c^	0.41 ± 0.03 ^a^	0.11 ± 0.05 ^c^	0	0	0.10 ± 0.04 ^c^
Luteinepoxide	0	0	11.51 ± 1.59 ^a^	2.00 ± 0.54 ^c^	0	0.20 ± 0.02 ^e^	3.62 ± 0.12 ^b^	1.10 ± 0.25 ^d^	10.11 ± 1.24 ^a^	0.73 ± 0.02 ^d^	0.31 ± 0.01 ^e^
Lutein	57.73 ± 1.81 ^a^	48.32 ± 2.85 ^b^	24.72 ± 2.75 ^e^	49.86 ± 2.44 ^b^	47.71 ± 2.54 ^b^	35.51 ± 2.85 ^d^	36.43 ± 2.52 ^d^	42.13 ± 2.62 ^c^	30.22 ± 2.42 ^e^	45.78 ± 2.14 ^bc^	43.97 ± 2.04 ^bc^
Zeaxanthin	0	0	13.21 ± 1.85 ^a^	2.02 ± 0.81 ^c^	0.31 ± 0.24 ^d^	0	1.61 ± 0.21 ^c^	0.60 ± 0.28 ^d^	9.43 ± 1.84 ^b^	1.62 ± 0.15 ^c^	0
Neolutein	0.50 ± 0.24 ^d^	9.51 ± 1.24 ^c^	7.42 ± 1.28 ^c^	9.91 ± 1.19 ^c^	18.11 ± 2.45 ^a^	13.50 ± 1.52 ^b^	0.40 ± 0.03 ^d^	9.51 ± 2.04 ^c^	10.90 ± 2.12 ^c^	13.42 ± 1.84 ^b^	9.52 ± 1.72 ^c^
α-carotene	trace	0	0	0	0	0	0	0	0	0	0
β-carotene	14.61 ± 1.08 ^a^	4.92 ± 1.22 ^c^	7.81 ± 1.32 ^b^	7.61 ± 2.01 ^b^	2.57 ± 0.58 ^d^	13.16 ± 2.12 ^a^	2.00 ± 0.24 ^d^	5.10 ± 1.24 ^c^	2.00 ± 0.95 ^d^	2.10 ± 0.93 ^d^	3.41 ± 0.95 ^d^

Different letters indicate significant differences (*p* < 0.05) between the experimental treatments for each carotenoid. The means ± standard errors, *n* = 3.

## Data Availability

All related data and methods are presented in this paper. Additional inquiries should be addressed to the corresponding author.

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
