# Peer review of "Post-Harvest Red- and Far-Red-Light Irradiation and Low Temperature Induce the Accumulation of Carotenoids, Capsaicinoids, and Ascorbic Acid in Capsicum annuum L. Green Pepper Fruit"

_foods, 2023, doi:10.3390/foods12081715_

Round 1

Reviewer 1 Report

The manuscript entitled “Post-harvest red, far-red light irradiation and low temperature induce the accumulation of carotenoids, capsaicinoids and ascorbic acid in Capsicum annuum L. Green Pepper Fruit” is an interesting contribution to understanding the impact of irradiation on carotenoids, capsaicinoids and ascorbic acid of one important fruits, as the Capsicum annuum. Brief comments and suggestions for improvement of manuscript as can be found as following:

In abstract:

The period time of irradiation that was applied in green fruit at both lights, for instance, in the second sentence “By briefly irradiating harvested fruit of Capsicum annuum L. hot peppers with red RL (660 nm) and far red light (FRL 730 nm), as well as keeping them at a low temperature for 24, 48 and 76 hours and we tried to determine whether the state of phytochromes in fruit affects 23 the biosynthesis of secondary metabolites.

The effects of irradiation treatment on total and carotenoids composition, the authors need to add the quantitative value that represents the increase of these components.

In Introduction:

Please add the word in the first sentence: “…fibre and their regular use…”

Line 44, before presented the scientific name of green pepper as C. annuum, firstly need appears as “Capsicum annum

Materials and methods

In plant materials more information regarding the numbers of plants or fruits that were subjected to irradiation needs to be mentioned. Also, the grow chamber had how much of diameter, high? The plants and the lamps were in the range of distanced? Please complemented this section of the relevant information. The assay was replicated? How many fruits?

In Line 131, add the reference number of Rodriguez-Amaya, 2001. The same at Line 134 “…previously (Ashikhmin et al., 2017)….”, Line 212 “…(Kreslavski et al., 2020).”

The legend of all figures should be moved to below of them. Please taking into account this along the manuscript.

The content of chlorophyll a and b, as well the carotenoids and all the chemical analysis that were performed resulting from how many determination? Please add this information in all the methodologies performed.

Results

In this section another quality parameters was evidenced but no mentioned until now and for this we suggested to add the information of water loss, how was done and the reference of methodology, should be add in the Materials and Methods sections.

Please rectify the word “caratoinoids” at table 1.

In Line 132 add the end point at the end of sentence.

The legend of Table 2 needs correction: the sentences should be moved to below of table: “Different letters indicate significant differences (p < 0.05) between the experimental treatments for each carotenoid. The means ± standard errors, n = 3.”

Author Response

In abstract:

  1. The period time of irradiation that was applied in green fruit at both lights, for instance, in the second sentence “By briefly irradiating harvested fruit of Capsicum annuum L. hot peppers with red RL (660 nm) and far red light (FRL 730 nm), as well as keeping them at a low temperature for 24, 48 and 76 hours and we tried to determine whether the state of phytochromes in fruit affects 23 the biosynthesis of secondary metabolites.

The effects of irradiation treatment on total and carotenoids composition, the authors need to add the quantitative value that represents the increase of these components.

Answer: We are grateful to the referee for the comment. Corrections in the abstract were made

In Introduction:

  1. Please add the word in the first sentence: “…fibre and their regular use…”

Answer: It is done.

  1. Line 44, before presented the scientific name of green pepper as C. annuum, firstly need appears as “Capsicum annum

Answer: It is done.

Materials and methods

  1. In plant materials more information regarding the numbers of plants or fruits that were subjected to irradiation needs to be mentioned. Also, the grow chamber had how much of diameter, high? The plants and the lamps were in the range of distanced? Please complemented this section of the relevant information. The assay was replicated? How many fruits?

Answer: It is done. This information was added to the materials and methods section «The LEDs were placed in fabric boxes with foil coating measuring 1.5 by 1.5 meters and 2 meters high. The lamps were located at a distance of 30 cm from the fruit, on both sides, and the fruit was placed on a grid in such a way as to avoid overheating of the fruit. The temperature was stabilized by the general climate system, as well as the exhaust fan system. In each version of the experiment, 20 pepper fruits were used with an average weight of about 50 g in the amount of about 1 kg.»

  1. In Line 131, add the reference number of Rodriguez-Amaya, 2001. The same at Line 134 “…previously (Ashikhmin et al., 2017)….”, Line 212 “…(Kreslavski et al., 2020).”

Answer: It is done.

  1. The legend of all figures should be moved to below of them. Please taking into account this along the manuscript.

Answer: It is done.

  1. The content of chlorophyll a and b, as well the carotenoids and all the chemical analysis that were performed resulting from how many determination? Please add this information in all the methodologies performed.

Answer: It is done.

 Results

  1. In this section another quality parameters was evidenced but no mentioned until now and for this we suggested to add the information of water loss, how was done and the reference of methodology, should be add in the Materials and Methods sections.

Answer: It is done. The text was added to Materials and Methods sections.

  1. Please rectify the word “caratoinoids” at table 1.

Answer: It is done.

  1. In Line 132 add the end point at the end of sentence.

Answer: It is done.

The legend of Table 2 needs correction: the sentences should be moved to below of table: “Different letters indicate significant differences (p < 0.05) between the experimental treatments for each carotenoid. The means ± standard errors, n = 3.”

Answer: It is done.

Reviewer 2 Report

It was honor to review manuscript “Post-harvest Red, Far-red Light Irradiation and Low Temperature Induce the Accumulation of Carotenoids, Capsaicinoids and Ascorbic Acid in Capsicum annuum L. Green Pepper Fruit” for FOODS journal.

All fruits, and pepper fruits are not exception, continue life after picking from plant. Usual postharvest treatment is cooling down on optimal temperature (7-10 °C) with goal to slow down metabolic processes, but to prevent development of chilling injuries. Despite best efforts, in stored fruits metabolic changes does occurs and it can be seen as color change (in case of green pepper – changes from green to yellow then orange and in majority of cultivars to red color). In presented manuscript effect of short storage (24h and 72h) on different temperatures (25 and 4°C) and effect of RL, FRL and combination of RL and FRL on green pepper fruits stored at 25°C on water loss, chlorophyll and carotenoid content, ascorbate and capsaicinoids content, photochemical activity and transcription levels of selected genes has been studied. As to been expected majority of changes after such short storage time are barely visible, however some of them are significant (for example ascorbic acid). Merit of manuscript is in transcript levels of selected enzymes which should explain changes of presented results. This contemporary concept is well designed, however execution of experiment as well as description of experiment in material and methods chapter combined with presentation of results and discussion of results is weak point of manuscript.

Pepper cultivar as well as use of cultivar should be stated (for fresh consumption, por processing, for paprika spice production…). Selection of genes monitored in experiment should be explained. There is also indication of possible mixed treatments: for example – FRL treated fruits had significantly higher water loss, which lead to question – are presented results consequence of higher temperature achieved by FRL treatment or consequence of higher temperature. Method MUST be described in detail since water loss after 24 for FRL and FL was higher than water loss after 72h. All results should be compared and contrasted to results available in literature (I believe it will be complicated since carotenoids in pepper are usually expressed per fresh weight). Carotenoids were determined without carotenoid standards (retention time and their spectral was used for determination) and they percentage was determined by using percentages of carotenoid area to the total area, so throughout text individual carotenoid should be referred as “percentage” not “content”. To total carotenoids (table 1) is referred to carotenoid content

Title of manuscript, goal (lines 106-109) and conclusion are not aligned. Please rewrite them so that they present meaningful sequence (if RL and FRL are mentioned in title, they should be also referred in in goal and in conclusion)!

LINES 44-46 Pepper fruit is well known as source of vitamin C. Please add vitamin C into sentence “The fruit of C. annuum are an important source of carotenoids and flavonoids that can slow the aging process and serve as a preventive measure for a number of diseases”. This is very important, since ascorbic acid is mentioned in title, it is part of results but there is no mentioning of vitamin C in introduction chapter!

LINES 46-47 “During storage, the pigments spontaneously degrade, which reduces the nutritional value of the fruit” This is only partially true. In pepper fruit chlorophyll level decrease during ripening while carotenoid content increases! Please find literature related to pepper pigments change during storage and rewrite sentence, with reference to nutritional value (also please refer to lines 439-443).

LINES 49-51. “This testifies in favor of the fact that environmental conditions can change the rate of accumulation or degradation of certain human useful nutrients.” – basically sentence is true, however beginning (“This testifies in favor”) where sentence relays on previous sentence is not correct since previous sentence states that carotenoids increases during ripening and maturing. Please rewrite and add references.

LINE 66 – “active form of phytochromes” use abbreviation “active form of PHYs“

Material and methods is weakest point of manuscript.

Lines 112-115. Which cultivar of pepper was used? What is the size of pepper sample used per treatment. THIS (sample size) is critical to be explained, since initial carotenoid content was 0,14, fruits treated with RL after 24h it increased carotenoid content to 0,51 and then after further 48h carotenoid content decreases to  0,32, without significant changes to percentage of individual carotenoid (Table 2). As someone who worked on biochemistry of carotenoids in pepper, such sudden increase looks more like laboratory error than sudden 3.6 fold carotenoid increase. Therefore, please explain sample size per treatment.

Line 117 – Instead of “Processing options” please use “light treatments”

Lines 117-121. Please be more precise about light treatments, especially treatment red light treatment (there is no description after “+25°Ð¡ light 660 nm 70 μmol (photons) m2 s-1” (lines 118-119). I suggest when describing treatments, use abbreviations which are already used in text. For example “Processing options used in the experiments were as follows: initial point - the beginning of the experiment (the same for all options); +4 °Ð¡ darkness (Dark 4 °Ð¡); +25 °Ð¡ darkness (Dark 25 °Ð¡); +25°Ð¡ light 660 nm 70 μmol (photons) m2 s-1 (RL); +25 °Ð¡ far-red light 730 nm 70 μmol (photons) 119 m2 s-1 (FRL); +25 °Ð¡ 660 nm + 730 nm 35 μmol (photons) m2 s-1 + 35 μm far-red light 730 nm 70 μmol (photons) m2 s-1 (RL+FRL) μmol (photons)….” Or any similar way which will better and more exact define treatments. Additionally throughout manuscript there is use of abbreviations FR and FRL – please use one of them. Due to effect of treatments of weight loss, please state temperature fruit during treatment and also time between treatment and every other parameter which could explain recorded difference in water loss!

Line 131 - Rodriguez-Amaya, 2001, referencing should be according to instructions for authors

Line 132 – “results were converted to g kg-1 of dry weight” – please ad, in brief method for dry matter

Line 134 - Ashikhmin et al., 2017, referencing should be according to instructions for authors

Lines 133-143 - HPLC analysis of carotenoids – there is no description of sample preparation (extraction of carotenoids from pepper fruit), which MUST be added. Please state sample size, how samples were homogenized, and procedure of carotenoid extraction (used solvents).

Lines 144-161. - HPLC analysis of ascorbate – there is no description of sample preparation (extraction of carotenoids from pepper fruit), which MUST be corrected. Please state sample size, how samples were homogenized, and procedure of extraction.

Line 162 - HPLC analysis of capsaicinoids – is method entirely developed by authors or some literature was used? If so, please cite

Line 210-234 RNA extraction and RT‒PCR.

It would be useful to mention the rationale for choosing target genes for transcriptional analysis. I suppose the choice was based due to their relationship in capsaicinoid biosynthesis. But, as we know, there is a wide set of related genes, so it might be interesting to emphasize specific criteria for selecting those exact genes. For example:

- genes controlling capsaicin accumulation (such as Pal)

- genes encoding the enzymes involved in fatty acid metabolism (such as Kas)

Specific details regarding PCR, number of cycles, fluorescent dye used, temperature and/or reference should be stated.

Perhaps all the details for target genes (PAL, CAM, SCY1, ACS, FATa, KAS, Act1), their forward and reverse primers, as well as their accession numbers can be presented in a table, to make them more presentable

Please provide more details about performed analyses (for example: whether it was used the comparative cycle threshold (∆∆Ct) method; automatically adjusted fluorescence threshold (ΔRn); and normalized to...; reference?)

Lines 235-243 Statistical data processing – since ANOVA was used to evaluate differences between treatments, please find a way to present significance of one-way ANOVA (for example add last row to tables and use **for p less than 0.01; *-for p 0.05-0.02 and N.S. if P is higher than 0.05).

Results: subject of RL and FRL and their effect on pepper fruit is contemporary subject in preset literature. Please compare effect of FRL and RL found in literature to your results.

Line 246-247 sentence “The influence of light of different quality and low temperature led to a decrease in the content of chlorophyll.” Is misleading. According to statistics, chlorophyll content is similar in treatments after 24 and 72h (all have letter b) with exception of RL+FR after 72h (letter c). So only RL+FR after 72h decreased chlorophyll content.

Table 1. All abbreviations (treatments, examined parameters) in the table should be listed and explained below the table, along with the units, where applicable. The table should have enough information to be interpreted independently, without consulting the manuscript. Please uniform far red light abbreviation trough manuscript (including tables FR or FRL). Water loss methodology MUST BE PRESENTED since water loss of RL+FR was 8,21 after 24h and after 72h moisture loss decreases to 5,48. Please make the table more systematic and transparent for interpreting the results; it is rather confusing and difficult to compare the obtained data among each other. Maybe adding some borders which separate 24 from 72h would increase clarity?

Lines 261-263 sentence “At the same time, the content of carotenoids increased mainly due to a decrease in the content of epoxy-lutein, zeaxanthin, and cis-mutatoxanthin, which was also accompanied by a decrease in the content of β-carotene (Table 2).” Please rewrite sentence, since decrease of percentage of epoxy-lutein, zeaxanthin, and cis-mutatoxanthin and β-carotene could not explain increase of carotenoid. In further sentence specify on which treatment is sentence realted to. Also, table 2 present percentages so word “content” shouldn’t be used (in relation to results presented in table 2, in order to use word content carotenoids must be determined in absolute units (mg/kg) and appropriate standards must be used).

Line 272, Table 2 Capsanthin second column (FRL, 24 h) 0.21±0.03 and last column (Dark 4°C, 72h) also 0.21±0.03 have different letter (b and c). Please recheck all lettering. In table heading, 72 hours it is written FR (and should be FRL). Also Table 2 is very hard to read so please make it more clear!! These things MUST be improved

a)      It MUST be clearly and visually noted on how long is 24(h) spread on (FRL, RL, RL+FRL, Dark 25, Dark 4C)

b)      I strongly suggest to present (in table 2) on separate page, using  landscape mode

c)       As in table 1, all abbreviations (treatments, examined parameters) in the table should be listed and explained below the table, along with the units, where applicable. The table should have enough information to be interpreted independently, without consulting the manuscript. Please uniform far red light abbreviation trough manuscript (including tables FR OR FRL).

d)      Why “Capsaxanthin” and “Capsarubin”? Please use capsanthin and capsorubin (as in conclusion (line 442)

e)      Please uniform far red light abbreviation trough manuscript (including tables FR OR FRL)

Lines 285-286 sentence “After 24 h exposure a similar increase in carotenoids was also observed under the FRL (31.3); however, it was only 2 times higher than in the control” The sentence is confusing! “Similar increase” (similar to what?) and nuber 31.3 does not appear in tables 1 and 2. Please rewrite!

Lines 286-288 sentence “The difference from RL option was only that the increase was due to other carotenoids, namely, violaxanthin, neoxanthin and lutein.” MUST be rewritten. Treatment of RL is compared to what?

Line 289 24°C or 25°C?

Lines 289-291 sentence “At 24 °C in the dark, the β-carotene content decreased compared to the initial point, probably due to an increase in neoxanthin content (Table 2).” MUST be rewritten. It is not “content” but “percentage” of β-carotene and secondly explanation of increase neoxanthin is not correct since neoxanthin percentage is same in initial point, 24h Dark 4°C and 72h Dark 4°C (all marked with b)!.

Line 293 “In the first 24 h, water loss was greatest in the RL+FRL and FRL variants while in the dark and on RL, dehydration was less pronounced“ according to my knowledge difference in water loss after 24h is rather experimental error. Was pepper fruit treated with RL+FRL and FRL treated latter than RL fruit. Was exposure to FRL heat up pepper fruits? This is very important since some responses (gene) of pepper fruit could attributed to higher temperature or to combination of temperature and FRL, instead of FRL alone.

Lines 305 – numbers 0.124 and 0.078 are not present in the table 1, please make correction

Line 307-308 please add at the end of sentence “when compared to initial value”

Line 308 and 309 control OR “initial value”

Line 310 please correct (Figure 2) to (Figure 2c)

Figure 2. Please uniform FR and FRL

Line 332 control OR “initial value”

Figure 3. all abbreviations (treatments, examined parameters) in the figure should be listed and explained below, along with the units, where applicable. The figure should have enough information to be interpreted independently, without consulting the manuscript. Please uniform far red light abbreviation trough manuscript (including tables FR OR FRL).

  Line 361 instead of “largest decrease” use “most significant decrease”

Line 374-377 such increase in carotenoid content in green pepper after 24h should be compared to literature data! Please find references and compare you findings!

Line 378 please replace content with percentage

Line 394 Pfr and Pf forms - It is not clearly defined what each of these forms means? activated vs. inactivated? please clarify this in the text by associating each form with the appropriate abbreviation so that it can be more easily followed through the manuscript. In general please take care on abbreviations throughout text

Lines 435-453 Conclusion – please state on which measured

Author Response

Reviewer 2

  1. Pepper cultivar as well as use of cultivar should be stated (for fresh consumption, por processing, for paprika spice production…).

Answer: It is done.

  1. Selection of genes monitored in experiment should be explained.

Answer: It is done. The text was added to the discussion section.

«…Capsaicinoids, the main secondary metabolites of pepper, are synthesized in two ways, the products of which are combined at the end of biosynthesis. The first pathway is phenylpropanoid, the key enzyme of which is also PAL, in which cinnamic acid is formed from phenylalanine, followed by cinnamate-4-hydroxylase (C4H), 4-coumarate-coenzyme A-ligase (4CL), hydroxycinnamoyltransferase (HCT), coumarate-3-hydroxylase (C3H), caffeyl-CoA-3-O-methyltransferase (CAM) and aminotransferase (pAMT) to form vanillamine…»

«…The work of Zhang with coauthors showed that in addition to PAL and capsaicin synthase (CAS), biosynthesis of fatty acids genes play a decisive role in the biosynthesis of capsaicinoids. The second pathway for the formation of capsaicinoids is associated with the biosynthesis of fatty acids. It includes the enzymes ketoacyl-ACP synthase I (KAS), malonyl-CoA-ACP transacylase (MCAT), acyl transfer protein (AСL), and FATa thioesterase (TE), among others [38,39]…»

  1. There is also indication of possible mixed treatments: for example – FRL treated fruits had significantly higher water loss, which lead to question – are presented results consequence of higher temperature achieved by FRL treatment or consequence of higher temperature. Method MUST be described in detail since water loss after 24 for FRL and FL was higher than water loss after 72h.

Answer: We are grateful to the referee for the question. The experiment was carried out in such a way as to exclude the possibility of heating the fruits, we tried to make the experiment one-factor. Necessary changes have been added to materials and methods.

  1. All results should be compared and contrasted to results available in literature (I believe it will be complicated since carotenoids in pepper are usually expressed per fresh weight).

Answer: It is done.

  1. Carotenoids were determined without carotenoid standards (retention time and their spectral was used for determination) and they percentage was determined by using percentages of carotenoid area to the total area, so throughout text individual carotenoid should be referred as “percentage” not “content”.

Answer: We are grateful to the referee for the question. In those places in the text where we discuss the qualitative composition, we have replaced the «content» with a «percentage».

  1. Title of manuscript, goal (lines 106-109) and conclusion are not aligned. Please rewrite them so that they present meaningful sequence (if RL and FRL are mentioned in title, they should be also referred in in goal and in conclusion)!

Answer: It is done.

  1. LINES 44-46 Pepper fruit is well known as source of vitamin C. Please add vitamin C into sentence “The fruit of C. annuum are an important source of carotenoids and flavonoids that can slow the aging process and serve as a preventive measure for a number of diseases”. This is very important, since ascorbic acid is mentioned in title, it is part of results but there is no mentioning of vitamin C in introduction chapter.

Answer: It is done.

  1. LINES 46-47 “During storage, the pigments spontaneously degrade, which reduces the nutritional value of the fruit” This is only partially true. In pepper fruit chlorophyll level decrease during ripening while carotenoid content increases! Please find literature related to pepper pigments change during storage and rewrite sentence, with reference to nutritional value.

Answer : The text was added: «During storage, the pigments in pepper fruits undergo changes, which can have varying effects on their nutritional value. While chlorophyll levels typically decrease during ripening, carotenoid content, such as beta-carotene and lutein, increases, contributing positively to the nutritional profile of the fruit. However, extended storage periods may lead to the degradation of certain pigments, potentially reducing the overall nutritional value of the pepper fruit.»

9 . LINES 49-51. “This testifies in favor of the fact that environmental conditions can change the rate of accumulation or degradation of certain human useful nutrients.” – basically sentence is true, however beginning (“This testifies in favor”) where sentence relays on previous sentence is not correct since previous sentence states that carotenoids increases during ripening and maturing. Please rewrite and add references.

Answer: The text was added to MS «Research indicates that environmental conditions can influence the rate of accumulation or degradation of essential nutrients, such as carotenoids, in fruits and vegetables during ripening and maturing processes»

Ripening improves the content of carotenoid, α-tocopherol, and polyunsaturated fatty acids in tomato (Solanum lycopersicum L.) fruits Ramesh Kumar Saini, Ahmad Jawid Zamany & Young-Soo Keum

Fruit ripening: dynamics and integrated analysis of carotenoids and anthocyanins Leepica Kapoor, Andrew J. Simkin, C. George Priya Doss & Ramamoorthy Siva BMC Plant Biology volume 22, Article number: 27 (2022)

  1. LINE 66 – “active form of phytochromes” use abbreviation “active form of PHYs“

Answer: It is done.

  1. Lines 112-115. Which cultivar of pepper was used? What is the size of pepper sample used per treatment. THIS (sample size) is critical to be explained, since initial carotenoid content was 0,14, fruits treated with RL after 24h it increased carotenoid content to 0,51 and then after further 48h carotenoid content decreases to 0,32, without significant changes to percentage of individual carotenoid (Table 2). As someone who worked on biochemistry of carotenoids in pepper, such sudden increase looks more like laboratory error than sudden 3.6 fold carotenoid increase. Therefore, please explain sample size per treatment.

Answer: We are grateful to the reviewer for the remark. The necessary changes have been added to the materials and methods section. We assume that during the first 24 hours, metabolic processes could indeed be carried out that enhance the biosynthesis of carotenoids, since the effect of the light was noticeable. The change in the percentage of carotenoids was also significant, for example, an increase in Cis-mutatoxanthin, Luteinepoxide Zeaxanthin was observed in red light in 24 h, and after 48 hours the percentage of these carotenoids decreased. We did not find experiments in which carotenoids were studied over time, on the other hand, in the article by Liu, C., Wan, H., Yang, Y., Ye, Q., Zhou, G., Wang, X., . .. & Cheng, Y. (2022). Post-Harvest LED Light Irradiation Affects Firmness, Bioactive Substances, and Amino Acid Compositions in Chili Pepper (Capsicum annum L.). Foods, 11(17), 2712. red light changes in total carotenoids were not significant, but we studied one point for 48 hours, which we did not study.

  1. Line 117 – Instead of “Processing options” please use “light treatments”

Answer: It is done.

  1. Lines 117-121. Please be more precise about light treatments, especially treatment red light treatment (there is no description after “+25°Ð¡ light 660 nm 70 μmol (photons) m2 s-1” (lines 118-119). I suggest when describing treatments, use abbreviations which are already used in text. For example “Processing options used in the experiments were as follows: initial point - the beginning of the experiment (the same for all options); +4 °Ð¡ darkness (Dark 4 °Ð¡); +25 °Ð¡ darkness (Dark 25 °Ð¡); +25°Ð¡ light 660 nm 70 μmol (photons) m2 s-1 (RL); +25 °Ð¡ far-red light 730 nm 70 μmol (photons) 119 m2 s-1 (FRL); +25 °Ð¡ 660 nm + 730 nm 35 μmol (photons) m2 s-1 + 35 μm far-red light 730 nm 70 μmol (photons) m2 s-1 (RL+FRL) μmol (photons)….” Or any similar way which will better and more exact define treatments. Additionally throughout manuscript there is use of abbreviations FR and FRL – please use one of them. Due to effect of treatments of weight loss, please state temperature fruit during treatment and also time between treatment and every other parameter which could explain recorded difference in water loss!

Answer: It is done. Materials and Methods Plant materials and experimental design secriot were totally revised.

  1. Line 131 - Rodriguez-Amaya, 2001, referencing should be according to instructions for authors

Answer: It is done.

  1. Line 132 – “results were converted to g kg-1 of dry weight” – please ad, in brief method for dry matter

Answer: It is done.

  1. Line 134 - Ashikhmin et al., 2017, referencing should be according to instructions for authors

Answer: It is done.

  1. Lines 133-143 - HPLC analysis of carotenoids – there is no description of sample preparation (extraction of carotenoids from pepper fruit), which MUST be added. Please state sample size, how samples were homogenized, and procedure of carotenoid extraction (used solvents).

Answer: Samples (200 mg) were homogenized in liquid nitrogen, and then 10 ml of 100% acetone was added to the resulting powder and stirred for 30 seconds and incubated for 5 minutes at room temperature. Then 4 ml of petroleum ether was added and repeated stirring. After that, 15-20 ml of distilled water was added and the upper fraction of petroleum ether was taken. Petroleum ether was dried under a stream of argon, then a film of carotenoids dissolved in 20 μl of acetone-methanol (7:2, V/V) and applied to an HPLC column.

  1. Lines 144-161. - HPLC analysis of ascorbate – there is no description of sample preparation (extraction of carotenoids from pepper fruit), which MUST be corrected. Please state sample size, how samples were homogenized, and procedure of extraction.

Answer: Extraction was carried out from 200 mg of an average sample. To do this, the powder ground in liquid nitrogen was immediately mixed with the extraction buffer of the following composition (74.448 mg EDTA; 286.65 mg Tris-(2-carboxyethyl)-phosphine hydrochloride (TCEP), 5 ml of 98% orthophosphoric acid Milli-Q water until 100 ml). The extraction buffer was prepared each time fresh immediately before analysis.

  1. Line 162 - HPLC analysis of capsaicinoids – is method entirely developed by authors or some literature was used? If so, please cite

Answer : The analysis was carried out by method Arce-Rodríguez, M.L., Ochoa-Alejo, N. Silencing AT3 gene reduces the expression of pAmt, BCAT, Kas, and Acl genes involved in capsaicinoid biosynthesis in chili pepper fruits. Biol Plant 59, 477–484 (2015). https://doi.org/10.1007/s10535-015-0525-y with some modifications.

  1. Line 210-234 RNA extraction and RT‒PCR.

It would be useful to mention the rationale for choosing target genes for transcriptional analysis. I suppose the choice was based due to their relationship in capsaicinoid biosynthesis. But, as we know, there is a wide set of related genes, so it might be interesting to emphasize specific criteria for selecting those exact genes. For example:

- genes controlling capsaicin accumulation (such as Pal)

- genes encoding the enzymes involved in fatty acid metabolism (such as Kas)

Answer: We are grateful to the referee for the question. For the synthesis of capsaicinoids, the work of the genes of phenylpropanoids and fatty acids is necessary, and then the products of these genes are combined into a capsaicin molecule. We have added a clarification to the discussion section

  1. Specific details regarding PCR, number of cycles, fluorescent dye used, temperature and/or reference should be stated.

Please provide more details about performed analyses (for example: whether it was used the comparative cycle threshold (∆∆Ct) method; automatically adjusted fluorescence threshold (ΔRn); and normalized to...; reference?)

Answer: Real-time PCR was performed using the SYBR ® Green qPCR supermixes for real-time PCR (Eurogen, Russia), the primers melting temperature was 60 °C, used the comparative cycle threshold (∆∆Ct) method. The transcript levels were normalized to the expression of the Actin1 gene. The level of gene transcripts at the initial point of the experiment was taken equal to 1.

  1. Lines 235-243 Statistical data processing – since ANOVA was used to evaluate differences between treatments, please find a way to present significance of one-way ANOVA (for example add last row to tables and use **for p less than 0.01; *-for p 0.05-0.02 and N.S. if P is higher than 0.05).

Answer: We thank the reviewer for the question. In tables and figures Different letters indicate significant differences (p < 0.05) between the experimental treatments for each column analysis. Due to the large amount of data and large tables, it is physically difficult to enter additional rows.

  1. Results: subject of RL and FRL and their effect on pepper fruit is contemporary subject in preset literature. Please compare effect of FRL and RL found in literature to your results.

Answer: It is known that different varieties of hot peppers can react differently to light of various spectral compositions. While the effect of red light on the biosynthesis of secondary metabolites has been previously studied, the impact of far-red light, as well as the combination of far-red light with red light, has not yet been sufficiently explored Liu C, Wan H, Yang Y, Ye Q, Zhou G, Wang X, Ahammed GJ, Cheng Y. Post-Harvest LED Light Irradiation Affects Firmness, Bioactive Substances, and Amino Acid Compositions in Chili Pepper (Capsicum annum L.). Foods. 2022 Sep 5;11(17):2712.

  1. Line 246-247 sentence “The influence of light of different quality and low temperature led to a decrease in the content of chlorophyll.” Is misleading. According to statistics, chlorophyll content is similar in treatments after 24 and 72h (all have letter b) with exception of RL+FR after 72h (letter c). So only RL+FR after 72h decreased chlorophyll content.

Answer: We rephrase the text «The influence of light of different quality and low temperature led to a decrease in the content of chlorophyll relative to the initial point of the experiment»

  1. Table 1. All abbreviations (treatments, examined parameters) in the table should be listed and explained below the table, along with the units, where applicable. The table should have enough information to be interpreted independently, without consulting the manuscript. Please uniform far red light abbreviation trough manuscript (including tables FR or FRL). Water loss methodology MUST BE PRESENTED since water loss of RL+FR was 8,21 after 24h and after 72h moisture loss decreases to 5,48. Please make the table more systematic and transparent for interpreting the results; it is rather confusing and difficult to compare the obtained data among each other. Maybe adding some borders which separate 24 from 72h would increase clarity?

Answer: The legend and structure of table 1 was improved «Chl - chlorophyll; Car + Xan carotenoids xanthophylls; Y(II) - effective quantum yield of PSII; Fv/Fm - maximum quantum yield of PSII. Different letters indicate significant differences (p < 0.05) between the experimental treatments. The means ± standard errors, n = 3.». The Water loss methodology was presented in the materials and methods section.

  1. Lines 261-263 sentence “At the same time, the content of carotenoids increased mainly due to a decrease in the content of epoxy-lutein, zeaxanthin, and cis-mutatoxanthin, which was also accompanied by a decrease in the content of β-carotene (Table 2).” Please rewrite sentence, since decrease of percentage of epoxy-lutein, zeaxanthin, and cis-mutatoxanthin and β-carotene could not explain increase of carotenoid. In further sentence specify on which treatment is sentence realted to. Also, table 2 present percentages so word “content” shouldn’t be used (in relation to results presented in table 2, in order to use word content carotenoids must be determined in absolute units (mg/kg) and appropriate standards must be used).

Answer: The text was rehrase « In variant RL+FRL the decrease in the percentage of epoxy-lutein, zeaxanthin, and cis-mutatoxanthin, which was also accompanied by a decrease in the percentage of β-carotene were observed»

  1. Line 272, Table 2 Capsanthin second column (FRL, 24 h) 0.21±0.03 and last column (Dark 4°C, 72h) also 0.21±0.03 have different letter (b and c). Please recheck all lettering. In table heading, 72 hours it is written FR (and should be FRL). Also Table 2 is very hard to read so please make it more clear!! These things MUST be improved. It MUST be clearly and visually noted on how long is 24(h) spread on (FRL, RL, RL+FRL, Dark 25, Dark 4C) I strongly suggest to present (in table 2) on separate page, using landscape mode

Answer: The tabe was improved. The letter of significant changes was improved.

  1. Lines 285-286 sentence “After 24 h exposure a similar increase in carotenoids was also observed under the FRL (31.3); however, it was only 2 times higher than in the control” The sentence is confusing! “Similar increase” (similar to what?) and nuber 31.3 does not appear in tables 1 and 2. Please rewrite!

Answer: It is done.

  1. Lines 286-288 sentence “The difference from RL option was only that the increase was due to other carotenoids, namely, violaxanthin, neoxanthin and lutein.” MUST be rewritten. Treatment of RL is compared to what? Line 289 24°C or 25°C?

Answer: It is done.

  1. Lines 289-291 sentence “At 24 °C in the dark, the β-carotene content decreased compared to the initial point, probably due to an increase in neoxanthin content (Table 2).” MUST be rewritten. It is not “content” but “percentage” of β-carotene and secondly explanation of increase neoxanthin is not correct since neoxanthin percentage is same in initial point, 24h Dark 4°C and 72h Dark 4°C (all marked with b)!.

Answer: It is done.

  1. Line 293 “In the first 24 h, water loss was greatest in the RL+FRL and FRL variants while in the dark and on RL, dehydration was less pronounced“ according to my knowledge difference in water loss after 24h is rather experimental error. Was pepper fruit treated with RL+FRL and FRL treated latter than RL fruit. Was exposure to FRL heat up pepper fruits? This is very important since some responses (gene) of pepper fruit could attributed to higher temperature or to combination of temperature and FRL, instead of FRL alone.

Answer: The fruits were irradiated in parallel in separate chambers, which were equipped with red, far red, as well as front and far red LEDs. Irradiation was carried out simultaneously from both sides and the distance to the fruit was about 30 cm, which practically excluded heating. The boxes were located inside a climate chamber equipped with a climate control system, and a ventilation system was installed in the boxes themselves.

The results obtained for different water losses are related to the different intensity of respiration and transpiration caused by different activation of photoreceptors.

  1. Lines 305 – numbers 0.124 and 0.078 are not present in the table 1, please make correction

Answer: It is done

  1. Line 307-308 please add at the end of sentence “when compared to initial value”

Answer: It is done

  1. Line 308 and 309 control OR “initial value”

Answer: It is done

  1. Line 310 please correct (Figure 2) to (Figure 2c)

Answer: It is done

  1. Figure 2. Please uniform FR and FRL

Answer: It is done

  1. Line 332 control OR “initial value”

Answer: It is done

  1. Figure 3. all abbreviations (treatments, examined parameters) in the figure should be listed and explained below, along with the units, where applicable. The figure should have enough information to be interpreted independently, without consulting the manuscript. Please uniform far red light abbreviation trough manuscript (including tables FR OR FRL).

Answer: It is done

  1. Line 361 instead of “largest decrease” use “most significant decrease”

 Answer: It is done

  1. Line 374-377 such increase in carotenoid content in green pepper after 24h should be compared to literature data! Please find references and compare you findings!

Answer: Similar results under RL conditions were obtained in the work «Kim, D., & Son, J. E. (2022). Adding far-red to red, blue supplemental light-emitting diode interlighting improved sweet pepper yield but attenuated carotenoid content. Frontiers in Plant Science, 13». At higher RL intensity on sweet peppers, on the contrary, a decrease in carotenoids in was observed (Naznin, M. T., Lefsrud, M., Gravel, V., & Azad, M. O. K. (2019). Blue light added with red LEDs enhance growth characteristics, pigments content, and antioxidant capacity in lettuce, spinach, kale, basil, and sweet pepper in a controlled environment. Plants, 8(4), 93.).

Line 378 please replace content with percentage

Answer: It is done

  1. Line 394 Pfr and Pf forms - It is not clearly defined what each of these forms means? activated vs. inactivated? please clarify this in the text by associating each form with the appropriate abbreviation so that it can be more easily followed through the manuscript. In general please take care on abbreviations throughout text

Answer: It is done.

  1. Lines 435-453 Conclusion – please state on which measured

Answer: It is done

Reviewer 3 Report

The authors of the manuscript titled " Post-harvest Red, Far-red Light Irradiation and Low Tempera ture Induce the Accumulation of Carotenoids, Capsaicinoids and Ascorbic Acid in Capsicum annuum L. Green Pepper Fruit” attempted to determine whether the state of phytochromes in fruit affects the biosynthesis of secondary metabolites. The topic is quite interest, and it fits into the scope of Foods. However, there are number of flaws in the experiment design and execution : 

1.      L96, L112, L115, What does technical ripeness mean? Do you have any reference about this?

2.      L112-113, Need to provide the manufacturer and country of grow chamber.

3.      L116, “from both sides”, How to change side? By hand or equipment?  Experimental design described not clear.

4.      Introduction needs to modify. Some sentences have grammar errors.

Author Response

  1. L96, L112, L115, What does technical ripeness mean? Do you have any reference about this?

Answer: Thanks for pointing out the mistake, the error was fixed.

  1. L112-113, Need to provide the manufacturer and country of the grow chamber.

Answer: Gorshkoff Prometheus series, Russia

  1. L116, “from both sides”, How to change side? By hand or equipment? Experimental design described not clear.

Answer: the experiment description was improved

  1. Introduction needs to modify. Some sentences have grammar errors.

Answer: Introduction section was improved

Reviewer 4 Report

This study discussed the effects of light with different spectral composition and temperature on the biosynthesis of secondary metabolites. Interesting information within a paper was presented.

1. Throughout the text, there are several acronyms that are not spelled out in full. It should be written in full the first time they are cited for readability.

2. Keywords: terms that already appear in the title are used. Number should be reduced to five keywords.

3. Figure 2: Image quality needs to be improved.

4. Please unify the citation format of references in the text

5. It would be useful if the manuscript provides HPLC chromatogram of samples and standards in the supplementary data in Table 2

6. Table 2 is confusing. Treatment for 24 h including which factors?

7. FRL is same with FR?

8. It would be useful if the manuscript provides key enzyme activity data, not just at the gene level.

Author Response

  1. Throughout the text, there are several acronyms that are not spelled out in full. It should be written in full the first time they are cited for readability.

Answer: It is done

  1. Keywords: terms that already appear in the title are used. Number should be reduced to five keywords.

Answer: It is done

  1. Figure 2: Image quality needs to be improved.

Answer: It is done

  1. Please unify the citation format of references in the text

Answer: It is done

  1. It would be useful if the manuscript provides HPLC chromatogram of samples and standards in the supplementary data in Table 2

Answer: It is done. Supplementary materials will be improved.

  1. Table 2 is confusing. Treatment for 24 h including which factors?

Answer: It is done

  1. FRL is same with FR?

Answer: We made the designation of far- red light uniform in the text

8 It would be useful if the manuscript provides key enzyme activity data, not just at the gene level.

Answer: We agree with the reviewer, it is very important to show the activity of enzymes, however, this is a very difficult work that can be devoted to a separate study.

Round 2

Reviewer 4 Report

Accept